# Investigation on Filaments for 3D Printing of Nasal Septum Cartilage Implant

**DOI:** 10.3390/ma16093534

**Published:** 2023-05-05

**Authors:** Przemysław Gnatowski, Karolina Gwizdała, Agnieszka Kurdyn, Andrzej Skorek, Ewa Augustin, Justyna Kucińska-Lipka

**Affiliations:** 1Department of Polymer Technology, Faculty of Chemistry, Gdańsk University of Technology, Gabriela Narutowicza Str. 11/12, 80-233 Gdańsk, Polandjustyna.kucinska-lipka@pg.edu.pl (J.K.-L.); 2Department of Pharmaceutical Technology and Biochemistry, Faculty of Chemistry, Gdańsk University of Technology, Gabriela Narutowicza Str. 11/12, 80-233 Gdańsk, Poland; 3Department of Otolaryngology, Faculty of Medicine, Medical University of Gdańsk, Marii Skłodowskiej-Curie Str. 3a, 80-210 Gdańsk, Poland

**Keywords:** 3D printing, nasal septum, nasal cartilage, deviated septum, polyurethane

## Abstract

Septoplasty is a widely used method in treating deviated septum. Although it is successfully implemented, there are problems with excessive bleeding, septal perforation, or infections. The use of anatomically shaped implants could help overcome these problems. This paper focuses on assessing the possibility of the usage of a nasal septum cartilage implant 3D printed from various market-available filaments. Five different types of laments were used, two of which claim to be suitable for medical use. A combination of modeling, mechanical (bending, compression), structural (FTIR), thermal (DSC, MFR), surface (contact angle), microscopic (optical), degradation (2 M HCl, 5 M NaOH, and 0.01 M PBS), printability, and cell viability (MTT) analyses allowed us to assess the suitability of materials for manufacturing implants. Bioflex had the most applicable properties among the tested materials, but despite the overall good performance, cell viability studies showed toxicity of the material in MTT test. The results of the study show that selected filaments were not suitable for nasal cartilage implants. The poor cell viability of Bioflex could be improved by surface modification. Further research on biocompatible elastic materials for 3D printing is needed either by the synthesis of new materials or by modifying existing ones.

## 1. Introduction

A deviated nasal septum can be a source of bleeding, nasal obstruction, and recurrent sinusitis. The ideal septum would be a simple structure, but most people have some degree of curvature or irregularity somewhere in the bony or cartilage portion. This becomes clinically significant when it leads to functional or aesthetic complications. Irregularities of the nasal septum cause twisting of the nose, humps, or dorsal depressions [1]. Effective surgical correction depends to a large extent on the accurate diagnosis of the anatomical point of the defect. As with any preoperative planning, the interview and physical examination are crucial to identify the cause of airway obstruction. Rhinoplasty planning is mainly based on the study of digital preoperative photographic images. An anterior rhinoscopy with a reflector is also used to visualize the nasal septum, as well as any disorders related to the structures located in its vicinity.

Septoplasty with reduction of the nasal septum is one of the most common surgical procedures to eliminate nasal obstruction [2]. A simple submucosal correction of bone or cartilage deviations is a technically uncomplicated and highly effective operation. Correction of the septum or other parts of the nose usually requires separation of the lateral tissues and placement of spreader grafts to open and maintain satisfactory angles, which prevents subsequent collapse during inspiration. It is essential to leave the dorsal support intact to maintain the stability of the middle and distal parts of the nose [2]. This technique most often consists of collecting cartilage and making grafts with dimensions of about 2–3 mm in width and 7–10 mm in length [3]. The best material for this type of procedure is autologous septal cartilage. It is firm and has excellent supportive properties to prevent deformation caused by skin contraction and scarring during the healing process. Unfortunately, its supply for direct transplantation is limited due to scarce tissue availability and solitary presence in the human body. In the case of too little living remnant of the septal cartilage, it is also possible to use the cartilage of the auricle and costal region, but there are not so many advantages when using them. Sampling of costal cartilage entails a higher risk of complications and is associated with a greater degree of resorption and distortion [4].

Fractures of the nasal bones often result in traumatic damage to the cartilage of the nasal septum coupled with severe secondary deformities [5]. Moreover, because surgery to correct a nasal fracture is relatively simple, accompanying nasal septal fractures are sometimes overlooked. To prevent secondary deformation, a bioabsorbable mesh is used as the inner splint. It is made using 3D printing technology using PLA [6] or PCL [7]. It has a simple design, thanks to which it can be easily inserted between the septal cartilage and the perichondrium without causing specific complications and without affecting the union. Depending on the patient’s metabolism, it degrades after two to three years. This is a sufficient period of time for the mesh to function satisfactorily as a support after the treatment. It is an effective procedure preventing secondary deformation of the cartilage, complementing the closed and open reduction performed in the case of fractures.

The treatment of nasal obstruction associated with internal valve obstruction and the collapse of the external nasal valve is possible through the use of an FDA-approved resorbable nasal implant. It consists of PLA, a 70:30 poly(L-lactide-co-D, L-lactide) blend, generally absorbed within 18–24 months after insertion. After this time, it is replaced by mature collagenous connective tissue, which is free from any inflammatory cells, with no signs of degradation of the PLA material [8].

The 3D printing process is based on the development of virtual three-dimensional plans of objects using a CAD file for computer design. It is then exported in STL form for slicer software, oriented in the most appropriate way for printing, divided into a number of layers, and finally combined to achieve the relevant shape. One of the printing technologies is the use of a specific instruction from the CAD system, which directs the print head along the x, y, and z planes. In this way, it is possible to build an object vertically, layer by layer. There are dozens of 3D printing processes. Each has a different technology platform, print accuracy, production efficiency, and input material requirements. However, using any of the technologies, it is possible to build a 3D object in whatever shape specified in the CAD file. Additive manufacturing is expected to revolutionize healthcare technology by providing advanced diagnostic and imaging options [9]. Since one of the modern goals of medicine is the implementation of personalized medicine, the 3D printing technique should support such approaches. With its use, it is possible to provide a product tailored to the needs of an individual patient in a short time without significantly affecting costs and without losing benefits [10]. Thee-dimensional printing techniques are highly versatile and are increasingly being used to produce custom and complex craniofacial components. By using appropriate materials for this purpose, such as polymers, it is possible to obtain structures that ensure the proper growth of new tissue or maintain the desired mechanical properties. However, there are still many limitations related to the biocompatibility and proper design of personalized implants.

Currently available prefabricated implants often require manual carving during the surgery, which is highly dependent on the surgeon’s skill. It can lead to sharp edges, which could injure the tissue, deviations in desired nose shape, or unstable positioning [11,12]. The usage of 3D printing in preparing nasal septum implants can overcome the mentioned problems [13]. The proposed technique could be used for the fabrication of biodegradable patient-specific implants with improved fit, comfort, and reduced complications. However, the patient-oriented design is only one of the many implant aspects to be considered. The other most important part is the selection of material, which implicates the mechanical and biological properties of the manufactured implant, which is in the scope of the presented paper.

This paper is dedicated to assessing the possibility of cost-effective nasal septum implant manufacturing by 3D printing from various market-available filaments. A nasal septum model was prepared, and a computer simulation of bending and compression was conducted based on the available literature data on nasal cartilage properties. Properties of five different filaments were analyzed by mechanical (bending, compression), structural (FTIR), thermal (DSC, MFR), surface (contact angle), microscopic (optical), degradation (2 M HCl, 5 M NaOH, and 0.01 M PBS), and printability studies and compared to cartilage properties. Based on this, the Bioflex filament was chosen for cell viability (MTT) analysis, which allowed us to assess the suitability of the material for manufacturing implants.

## 2. Materials and Methods

### 2.1. Filaments

The study was performed on 5 different filaments: thermoplastic poly(ester urethane) (TPU) (TPU, DevilDesign, Mikołów, Poland), MediFlex (MediFlex 96, Noctuo, Gliwice, Poland), polylactide (PLA) (Easy PLA, Fiberlogy, Brzezie, Poland), poly-ε-caprolactone (PCL) (PCL, SUNLU, Zhuhai, China), and Bioflex (Bioflex, Filoalfa, Ozzero, Italy). TPU was chosen due to its excellent flexibility. MediFlex was chosen because it is made from raw materials of high medical grade. PLA and PCL filaments were chosen because both materials are most commonly used to produce nasal septum scaffolds. Bioflex was chosen due to its flexibility and medical certification. The summary of properties of the chosen filaments is presented in Table 1.

### 2.2. Anatomical Model Design

The septal cartilage model was prepared using an MRI scan of the human head of one male adult subject without septal deviation. The segmentation process was achieved using 3D Slicer 5.2.1 software, which is an open-source software platform for visualization and medical image computing [19]. By a combination of threshold (43 to 95), scissors, islands (remove smaller than 500 voxels), and smoothing (kernel size 2 mm, 5 mm, and 10 mm) tools, the septal cartilage model was prepared. The obtained model was exported from the 3D Slicer program in the form of STL files. Autodesk Inventor Professional 2023 software was used to process the exported STL files and prepare models suitable for the study of degradation and mechanical analysis.

### 2.3. Stress Analysis

The study consisted of simulating the bending and compressing of the human cartilage tissue, which corresponded to the designed implant, using Stress Analysis tool in Autodesk Inventor Professional 2023 software. Based on literature data corresponding to the properties of cartilage, a new material was defined and stored in the Inventor material library. The material properties applied to human cartilage are presented in Table 2. Next, the newly defined material was assigned to the prepared model of cartilage. For the need to determine the immovable elements necessary for static analysis, the implant was modified by adding supports. Loads of different ranges were applied to the model surface.

### 2.4. Fourier Transform Infrared Spectroscopy (FTIR)

The FTIR Nicolet iS 10 spectrometer (Thermo Fisher Scientific, Waltham, MA, USA) was used to examine the chemical functional groups of the tested filaments. The spectral range was from 4000 to 500 cm^−1^, using 16 scans with a resolution of 4 cm^−1^.

### 2.5. Differential Scanning Calorimetry (DSC)

The DSC measurements were performed using a Netzsch 204F1 Pheonix apparatus with Proteus80 software (Netzsch, Selb, Germany) under a nitrogen atmosphere (flow rate 20 mL min^−1^) at the temperature range of 25/30–250 °C. The heating/cooling rate was 10 K min^−1^.

### 2.6. Melt Flow Rate (MFR) and Melt Volume Rate (MVR)

The MFR and MVR of the studied filaments were measured using load plastometer Zwick Roell BMF-001 and TestXpert^®^ II software. The study conditions were based on ISO 1133 standard. The measurement was conducted at printing temperatures (TPU, PLA–215 °C, MediFlex–240 °C, Bioflex–190 °C, and PCL–180 °C) under a load of 2.16 kg. Three repetitions were performed for each material, and the result was presented as an average (n = 3).

### 2.7. Three-Dimensional Printing

Original Prusa i3 MK3S+ (Prusa Research, Prague, Czech Republic), an FFF 3D printer with PrusaSlicer (2.5.0 version) was used to print the test elements. The most important 3D printing parameters of all test samples are listed in Table 3. Parameters were established based on previous DSC and MFR tests.

### 2.8. Contact Angle

Contact angle study was performed using a Rame-Hart 90-U3-PRO goniometer and dedicated software Drop Image Pro. Prior to measurements, pieces of filament were pressed on a hot plate heated to extrusion temperature (Table 3) to obtain thin, flat films. Ten measurements were taken for each sample, which allowed for the calculation of average values. The liquid used in the analysis was water. The test material was placed on the goniometer table. Then, using a syringe, a drop of water was applied to the structure and measured using a camera mounted in front of the table. Completed results for each sample in the form of the contact angle, work of adhesion, or surface energy were obtained using dedicated software.

### 2.9. Compression and Flexural Strength/Mechanical Properties

Compression strength was performed on the Zwick & Roell Z020 machine and compatible TestXpert^®^ II software. Samples for the compressive strength testing were prepared according to ISO 604 standards. The samples were in the shape of cuboids with dimensions of 10 × 10 × 4 mm. Five samples were printed for each material with infill along the compression direction and five samples with infill transverse to the compression direction. Initial force (F_0_) was 5 N with an initial crosshead speed (V_0_) of 50 mm/s. Crosshead speed during the test (V) was 1 mm/min. The measurement was conducted up to the maximum length change of 60%.

For flexural strength, a three-point bending test was conducted on the Zwick & Roell Z020 machine and compatible TestXpert^®^ II software. Flexural strength tests were performed on 3D-printed samples of the nasal septum with added supports. The parameters of the test were based on ISO 178 standard. The study was carried out at room temperature. Distance between supports (L) was 13 mm, and initial force (F_0_) was 1 N with initial crosshead speed (V_0_) of 50 mm/min. Crosshead speed during the test (V) was 60 mm/min. The measurement was conducted up to the maximum length change of 6 mm.

### 2.10. Degradation Studies

The degradation studies of filaments were conducted in selected media: 2 M HCl (pH = ~0), 5 M NaOH (pH = ~14), and 0.01 M PBS (phosphate-buffered saline) (pH = 7.4). The medium was replenished every 7 days [24]. Before replenishing each medium, the pH of solutions was checked (PH-100 ATC, Voltcraft) and corrected if needed. For the study, disc-shaped samples with a height of 3 mm and a diameter of 13 mm were used. The obtained samples were cleaned in ethanol and then dried in a thermobalance (RADWAG MAX50/SX) set at 40 °C and weighed on an analytical balance (RADWAG AS 220.R2). Nine discs were printed from each material, three discs per medium, and then placed in 24-well cell culture plates filled with 2.5 mL of degradation medium. The samples were incubated at 37 °C for 56 days. The weight change in the samples was examined after 1, 2, 7, 14, 21, 28, 42, 49, and 56 days of incubation in the media. The mass change in the samples was measured as follows: Samples were removed from a 24-well plate and dried on a paper towel to remove excess solution. The samples were then placed in a thermobalance for complete drying and weighed. Mass loss was calculated by Formula (1):(1)$$∆m=\left(\frac{m_{0}-m_{1}}{m_{0}}\right)\times 100\%,$$
where *m*_0_ is the sample weight before the test (g) and *m*_1_ is the sample weight incubation (g). Three samples were tested for each medium, and the result was presented as an average (n = 3).

### 2.11. Cell Culture, Cytotoxicity, and Morphology Assesment

The cytotoxicity of Bioflex filament was studied according to the ISO 10993-5 standard in indirect cytotoxicity test using MTT assay. Only Bioflex printouts were tested due to favorable properties of the material, i.e., similar mechanical properties of the implant in comparison to septum cartilage and lowest contact angle of all filaments.

Mouse fibroblast L-929 cells (Sigma Aldrich/Merck, St. Louis, MO, USA) were chosen due to being recommended for the biological evaluation of biomedical devices in ISO 10993-5 standard. L-929 cells were cultured in medium recommended by manufacturer: low-glucose Dulbecco’s modified Eagle’s medium (DMEM LG, Sigma Aldrich/Merck, St. Louis, MO, USA) supplemented with 10% heat-inactivated fetal bovine serum (FBS; Biowest, Nuaille, France), 100 µg/mL streptomycin, 100 U/mL penicillin, and 2 mM glutamine (Sigma Aldrich/Merck, St. Louis, MO, USA). Cells were incubated in a humidified atmosphere containing 5% CO_2_ at 37 °C. All the experiments were carried out with cells in the exponential phase of growth.

Four samples were printed according to paragraph 2.7. in the shape of a 46 × 46 × 1 mm cuboid. Printouts were sterilized for 30 min each side under UV radiation. Then, 70 mL of extract of the studied samples was prepared in culture medium DMEM LG supplemented with 10% FBS, 100 U/mL penicillin, 100 µg/mL streptomycin, and 2 mM glutamine. The prepared extract was incubated for 24 h at 37 °C and 5% CO_2_.

The L-929 cells were seeded in 24-well plates with a density of 20,000 cells/well and cultured for 24 h to allow for attachment. Then, cell culture medium was removed and replaced with extract solutions for the next 24 h, 48 h, and 72 h. DMEM LG supplemented with FBS and antibiotics was used as a non-toxic control. After the cell incubation with extract solutions, the MTT assay was performed. The absorbance of the prepared solutions was measured at λ = 540 nm using iMark Microplate Absorbance Reader (Bio-Rad, Hercules, CA, USA). The results from three independent experiments (n = 3) were shown in the graph as cells’ viability towards control (100% of viability). The statistical analysis was performed with the use of Origin Pro 9.0. Statistical differences were evaluated by the one-way ANOVA (α = 0.05) and Tukey’s post hoc test (α = 0.05).

The morphology of L-929 cells following treatment with the extract solutions was evaluated using light microscope (Olympus IX83, Tokyo, Japan; objective 100× magnification).

## 3. Results

### 3.1. Anatomical Model Design

The preparation of the anatomical model of the nasal septum implant started with a segmentation process using 3D Slicer. The obtained nasal septum model is presented in Figure 1. The model was then exported to STL format and measured in Autodesk Inventor Professional software. Based on the measurements, a sketch was prepared (Figure 2a) and then extruded (Figure 3a). The model was modified to prepare the implant for bending testing by adding the supports (Figure 2b). The comparison between human nasal septum and the proposed implant is presented in Figure 3b. The proposed implant is around 10% smaller in comparison to the septum to allow easy placement and anchorage to the tissue by the doctors. If needed, the implant could be cropped by the surgeons. The proposed procedure for implant preparation is time-efficient and does not require specialist training, thus it can be easily implemented for the preparation of patient-specific implants.

### 3.2. Stress Analysis

The results of the stress analysis of the prepared models with the properties of human septum cartilage in Autodesk Inventor Professional 2023 are presented in Table 4. The graphical interpretation of the simulation is presented in Figure 4. The results were linear in both cases, and the warning messages about scalability of the study were displayed at the force of 7 N (for bending) and 90 N (for compression). The obtained modulus in compression simulation was 2.28 MPa, which is similar to the results of human nasal cartilage reported in the literature [20], thus it can be concluded that the stress analysis results are valid for further comparison with tested filaments.

### 3.3. Fourier Transform Infrared Spectroscopy (FTIR)

The FTIR spectra of studied filaments are presented in Figure 5 and the descriptions of characteristic bands for the studied filaments are summarized in Table 5. In the spectra of all studied filaments, medium signals between 3000 cm^−1^ and 2800 cm^−1^ could be attributed to symmetric and asymmetric stretching vibrations of C-H bonds from methylene and methyl groups. In the case of TPU, PLA, and PCL, strong signals between 1750 cm^−1^ and 1725 cm^−1^ could be attributed to C=O stretching vibrations of ester groups and strong signals at around 1160 cm^−1^ could be attributed to stretching vibrations of C-O bonds [25,26]. TPU and Bioflex filaments also exhibited strong signals at around 1690 cm^−1^ and 1220 cm^−1^, characteristic of vibrations of C=O and C-N bonds from urethane groups [27,28]. Additionally, Bioflex showed a very small signal and TPU filament showed a medium signal at around 3300 cm^−1^, which could be attributed to stretching vibrations of N-H bonds. In the case of Bioflex, a signal at around 1100 cm^−1^ could be attributed to stretching vibrations of C-O ether groups [29]. Based on these findings, it could be said that the TPU filament was based on polyester polyurethane and Bioflex was based on polyether polyurethane. Based on the FTIR analysis MediFlex filament’s chemical structure could not be revealed. Visible aromatic signals are present in the form of peaks at around 1600, 1490, and 1450 cm^−1^, which could be assigned to stretching vibrations of aromatic C=C bonds, and strong signals at 740 and 700 cm^−1^, which could be attributed to out-of-plane C-H bending bond vibrations. There is also a tirade signal present at the 3100–3000 cm^−1^ range, which could be attributed to C-H stretching vibrations of aromatic bonds and benzene fingers at the 2000–1500 cm^−1^ range [30]. It could be concluded that MediFlex is mainly composed of aromatic-based hydrocarbon polymers, e.g., polystyrene.

### 3.4. Differential Scanning Calorimetry (DSC)

The DSC curves are presented in Figure 6 and the observed transition peaks are described in Table 6. The results correspond to findings from the FTIR study. The Bioflex filament showed a very broad peak between 105 and 140 °C, characteristic of polyether-based polyurethanes [31]. The TPU filament showed the melting of hard segments at the range of 205–225 °C, characteristic of poly(ester-urethane) based on aromatic diisocyanates [32]. MediFlex exhibited very small transition energy at the 92–99 °C range, which is characteristic of polystyrene [33]. The PCL filament showed a strong peak at the 55–64 °C range, which is typical of PCL polymers. The second transition temperature at 75–91 °C could be present due to some impurities, e.g., PVC [34]. The PVC impurity could also be confirmed with FTIR spectra (Figure 5) with a signal around 640 cm^−1^, which could be assigned to stretching vibrations of the C-Cl bond [35]. PLA thermogram shows typical behavior of a PLA polymer: glass transition in the range of 59–62 °C, broad cold crystallization area in the range of 109–139 °C, and melting transition at 146–156 °C [36]. It is worth mentioning that all observed transitions were lower than the recommended printing temperatures, with the exception of TPU, where the supplier recommends a printing temperature of 210 °C. Higher printing temperatures are probably recommended to increase the flow rate of polymers.

### 3.5. Melt Flow Rate (MFR) and Melt Volume Rate (MVR)

The summary of MFR and MVR is presented in Figure 7 and Table 7. At the temperature of printing, the highest flow was observed for polyurethane-based filaments: Bioflex and TPU (70.20 ± 0.67, 51.11 ± 0.93 g/10 min and 52.57 ± 0.50, 47.3 ± 1.1 cm^3^/10 min, respectively). The PCL filament also flowed well (37.13 ± 0.81 g/10 min and 34.2 ± 1.3 cm^3^/10 min). The lowest flow rates were observed for PLA and MediFlex (16.59 ± 0.36, 10.25 ± 0.31 g/10 min and 15.01 ± 0.31, 11.67 ± 0.3 cm^3^/10 min, respectively). For 3D printing, application flow rates should not be too high, to maintain dimensional stability and prevent uncontrolled material leaks, but also not too low, to allow quick plasticization and the appropriate amount of extruded material with decent printing speeds [37,38]. Usually, an MFR of around 10 g/10 min is considered the lowest applicable in 3D printing [39], and all proposed filaments exhibited higher values.

### 3.6. Contact Angle

The results of the contact angle study are presented in Figure 8. Only two materials exhibited hydrophilic behavior, namely Bioflex and PLA (61.5 ± 2.2 and 75.52 ± 0.97° at start). Other materials had contact angles over 90°. Generally, a contact angle in the range of 50 to 70° coupled with high surface energies enhances the adsorption of cells and proteins, thus having a beneficial effect on integration between implants and tissues [40,41,42]. Higher hydrophilicity (lower contact angle) can disturb the interactions between cells sown on the implant surface, and a higher contact angle (higher hydrophobicity) can result in lowered cell adhesion to the printout, reducing biocompatibility as a result. Adhesion of the cells to the implant surface is mediated by the adhesion of the proteins [43]. It is known that different proteins react in distinct ways depending on the wettability of the surface [44,45]. Recent studies suggest that hydrophilic surfaces could benefit anti-inflammatory cytokine production, while hydrophobic surfaces promote pro-inflammatory cytokine generation [46,47]. The presented findings indicate that PLA and Bioflex could be considered as materials for producing implants due to their hydrophilic behavior with moderate wettability.

### 3.7. Compression and Flexural Strength/Mechanical Properties

The results of mechanical testing are presented in Table 8 and Figure 9, Figure 10 and Figure 11. All prepared samples exhibited higher mechanical properties compared to simulated nasal septum cartilage properties. In all tests, the closest properties to native tissue were exhibited by samples prepared from Bioflex material, especially in low-strain regions (up to 10% in compression test and up to 1 mm in bending test). Another material with close properties was TPU. The implant’s mechanical properties should be as close as possible to native tissue to ensure uniform distribution of stress at the implant and good stress transfer from the attached implant to the tissue, which results in smaller movements at the implant–tissue interface, thus reducing healing times [48,49]. What is worth mentioning is that the infill direction has an impact on the compression behavior of samples. Samples with the infill printed longitudinally to the stress direction had higher (TPU, PCL, and PLA) moduli, which increases the stiffness at lower strains. On the other hand, samples with the infill printed transversally to the stress direction showed higher stiffness at larger strains, resulting in 38 to 91% higher stresses at 60% of deformation in comparison to longitudinally printed samples. Comparing to reported results in the literature on clinically applied PCL nasal septum implants, Bioflex offers similar bending strength (PCL 24.10 ± 1.06 MPa, 28.9 ± 1.3 N [50]), which makes it a possible alternative to PCL-based implants in this regard.

### 3.8. Degradation Studies

A summary of the in vitro degradation studies is presented in Figure 12. High concentrations of acidic and alkaline solutions allowed the time of examination to be reduced in comparison to traditional studies in saline [51,52]. The only material which exhibited substantial mass losses in all three mediums was BioFlex. The mass loss on average was the highest in 5 M NaOH, but in the range of the measuring error of 2 M HCl (31.5 ± 8.7% and 21.2 ± 6.0% of remaining mass after 56 days in 2 M HCl and 5 M NaOH, respectively). The mass loss in 0.01 M PBS solution was low, but noticeable (92.7 ± 5.2% remaining mass after 56 days). The results are comparable to previous studies on this material [37]. The TPU material showed degradation in both HCl and NaOH (14 ± 8.3% and 58.50 ± 0.50% remaining mass after 56 days, respectively). The susceptibility of TPU to acidic degradation is reported in the literature [53]. It is worth mentioning that until the 28th day, the degradation rate in HCl and NaOH was similar, then it accelerated in HCl, while in NaOH the mass loss was linear. In the case of MediFlex, no degradation was observed. PCL was also susceptible to degradation in NaOH (complete degradation after 49 days) and in HCl (55.0 ± 2.9% remaining mass after 56 days), which is known in the literature [51,54]. PLA showed complete degradation in NaOH after just 2 days, while being intact in other media. Such behavior was expected, as PLA is known for being susceptible to hydrolytic degradation [55]. Discussing the degradation patterns, it is essential to underline that depending on the place in the body and the type of implant (support or structural), the behavior will be different. In the case of supporting implants attached to the septum, even 10 to 20% of mass loss can result in detachment of the implant and expulsion from the body. In the case of structural implants, higher degrees of mass loss will result in material disappearance. The degradation pattern of the material selected for use as an implant should allow for the full recovery of tissue functions during implant service. If an implant degrades too fast, the restored tissue could be misshapen or hollow. On the other hand, if an implant stays in place for too long, it can hinder the healing process or lead to tissue overgrowths.

### 3.9. Cytotoxicity Studies

Based on previous studies, the Bioflex filament was chosen for cytotoxicity studies, as it had the best mechanical fit to nasal septum cartilage, the lowest contact angle, and appropriate degradation behavior for a long-term implant. The results of these studies are shown as the cell viability following the extract treatment compared to the control, cultured in medium without extract (Figure 13). Additionally, pictures presenting the morphology of L-929 cells are shown in Figure 14. The positive result of that in vitro test could indicate the possible biocompatibility in vivo. Such a relation could be seen for 80 kDa PCL, which tested positively for MTT [56,57] and then was successfully tested on a 3D porcine nasal septal cartilage in vitro model [58] and in vivo as nasal dorsum augmentation in minipigs [59].

According to ISO 10993-5 standards, the reduction in cell viability by more than 30% is considered a cytotoxic effect. After 24 h of incubation, 100% extract exhibited moderate cytotoxicity, while lower concentrations did not. After 48 and 72 h, only 12.5% of the extract was considered biocompatible, with cell viability of around 80%. In turn, 25 and 50% extracts could be described as moderately cytotoxic following 48 and 72 h of incubation. The morphological examination of L-929 cells incubated with the studied extracts confirmed the cytotoxicity results. Cells treated with 12.5% extract for 24, 48, and 72 h presented unchanged morphology compared to the control, untreated cells. In contrast, treatment with 100% extract for 48 and 72 h resulted in morphological changes. Cells were smaller, shrunken, and they detached from the culture plate, which may indicate that these cells have died. Haryńska et al. report the full biocompatibility of the Bioflex filament in the same extract concentrations after 24 h [37] using CCl 163 cells. The main reasons for the discrepancy between the presented results and the literature could be the higher material contact surface or different cells used. In this study, gyroidal infill of 20% was used, whereas Haryńska et al. used 85% triangular infill, thus the samples used here had a higher contact area, which could result in higher cytotoxicity. Additionally, the volume of extract was not disclosed, which could have an effect on the overall cytotoxicity result. Based on the study results, the Bioflex filament used standalone is not suitable for usage as an implant; however, the biocompatibility of the proposed implant could be improved, e.g., by applying a functional hydrogel coating, deposition of biological molecules, or wet surface modification with active substances [60]. It is also possible to use Bioflex as a base for blends with more biocompatible materials, such as the previously mentioned PCL. Proper biological interactions are fundamental for a regenerative effect and the prevention of inflammatory reactions. High biocompatibility is essential for admission to clinical trials, thus, despite the overall suitable properties of the proposed implant, the cell viability has to first be improved.

## 4. Conclusions

Based on mechanical, degradation, and contact angle studies, among the tested materials, the best for the proposed nasal septum cartilage implant was Bioflex. It had the closest mechanical properties to native tissue, especially in low-strain regions, a stable degradation rate, and a low contact angle. Cytotoxicity studies showed that this material maintained biocompatibility only at a low concentration (12.5%) and after a short incubation time (24 h). Following a longer incubation time (48–72 h), Bioflex was cytotoxic, especially at 100% concentration. For the proposed nasal septum implant, other elastic materials with similar properties, but higher biocompatibility, have to be tested. There are not many other commercially available elastic filaments with medical certification. Thus, it could be necessary to synthesize a new material with tunable properties to fill the requirements of nasal septum implants. Alternatively, Bioflex could be used after improvement of its biocompatibility, e.g., by surface modifications or by blending it with other cytocompatible materials. Although the paper focuses on nasal septum implants, the presented results may be useful in the preparation of other nasal cartilage implants, such as alar or lateral implants.

## Figures and Tables

**Figure 1 materials-16-03534-f001:**
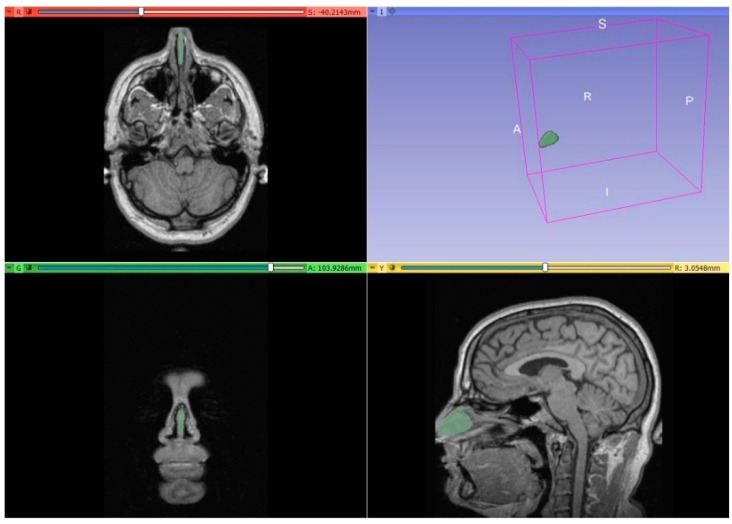
Separated nasal septum based on MRI scan in the 3D Slicer program.

**Figure 2 materials-16-03534-f002:**
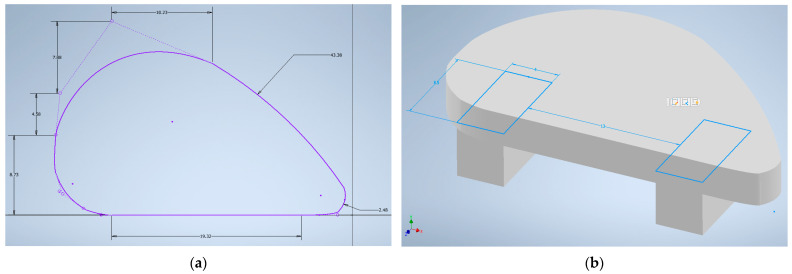
Sketches in Autodesk Inventor 2023: (**a**) 2D sketch of nasal septum implant based on obtained STL model from 3D Slicer program; (**b**) sample for bending testing with support height of 4 mm.

**Figure 3 materials-16-03534-f003:**
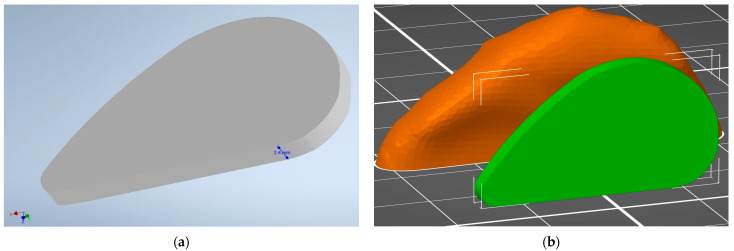
Three-dimensional model of nasal septum implant obtained in Autodesk Inventor Professional 2023: (**a**) general view of septum implant in Autodesk Inventor 2023; (**b**) comparison in PrusaSlicer of prepared implant model (green) to STL model of septum (orange) obtained from 3D Slicer program. The visible grid size is 10 × 10 mm.

**Figure 4 materials-16-03534-f004:**
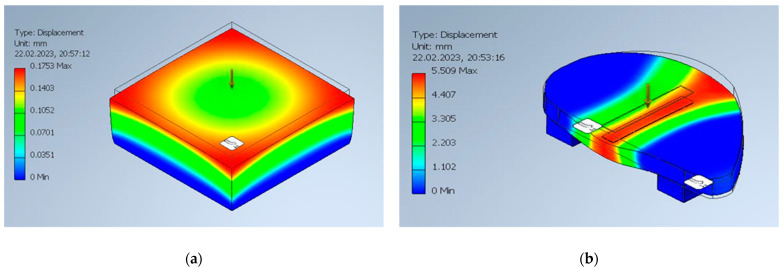
Graphical interpretation of mechanical simulation results of nasal septum cartilage: (**a**) compression study at 10 N of force; (**b**) bending study at 8 N of force.

**Figure 5 materials-16-03534-f005:**
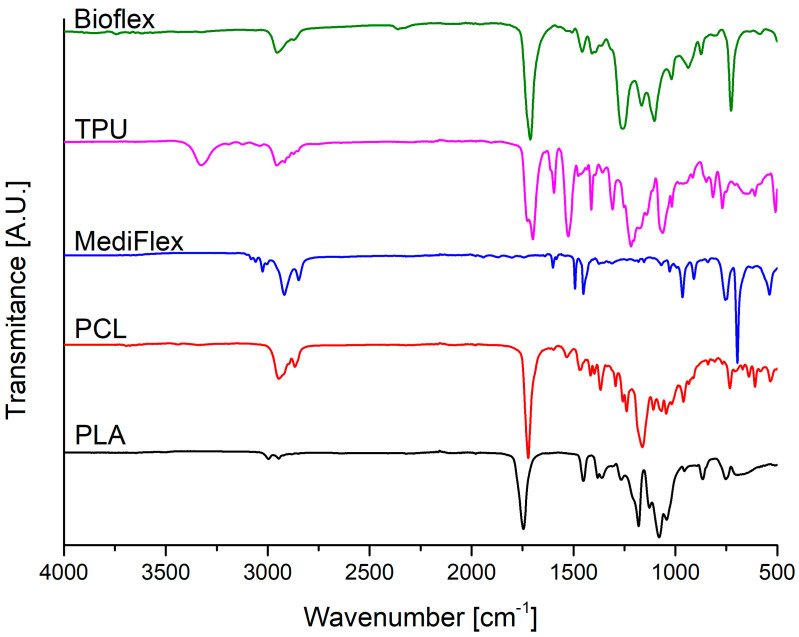
FTIR spectra of filaments.

**Figure 6 materials-16-03534-f006:**
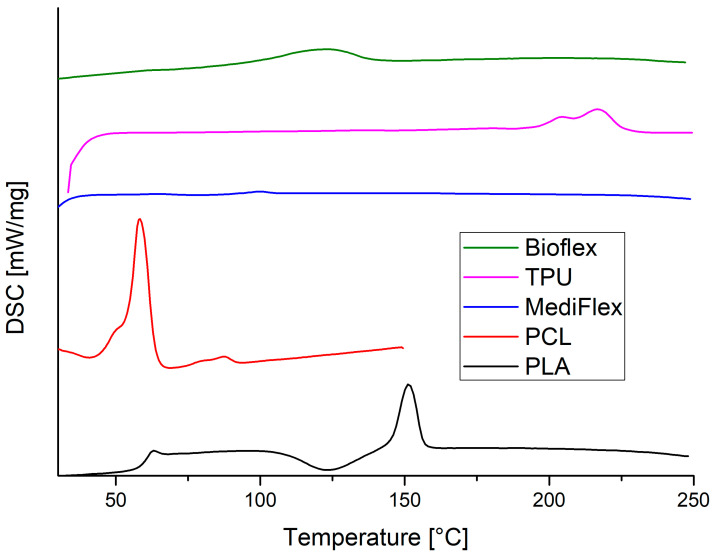
Second heating DSC curves of filaments.

**Figure 7 materials-16-03534-f007:**
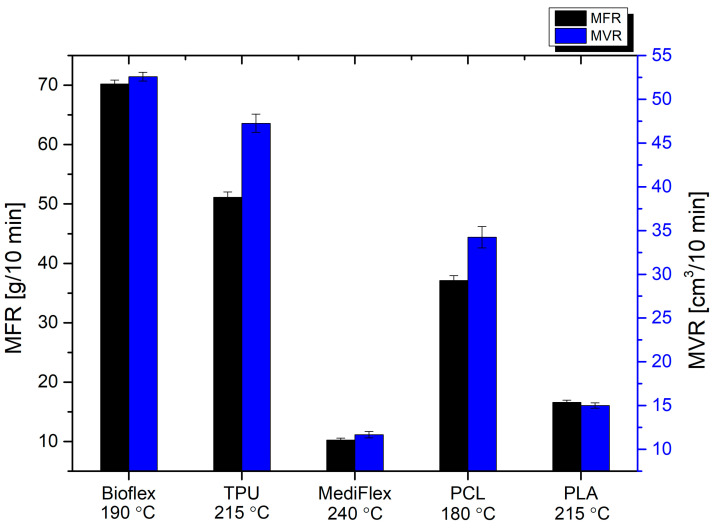
MFR and MVR of filaments.

**Figure 8 materials-16-03534-f008:**
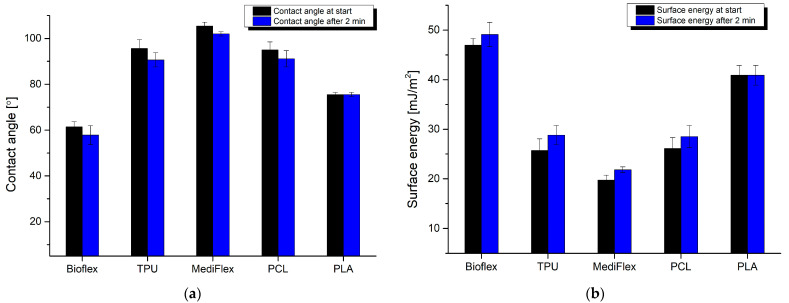
Results of filaments’ contact angle measurement: (**a**) water contact angle; (**b**) surface energy.

**Figure 9 materials-16-03534-f009:**
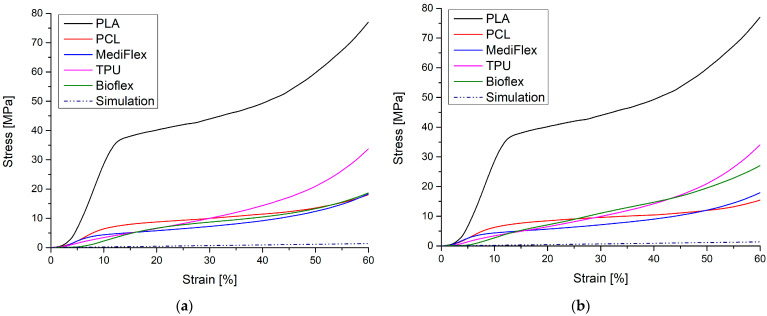
Exemplary compression stress–strain curves for tested filaments and simulated nasal cartilage behavior: (**a**) longitudinal measurements; (**b**) transversal measurements.

**Figure 10 materials-16-03534-f010:**
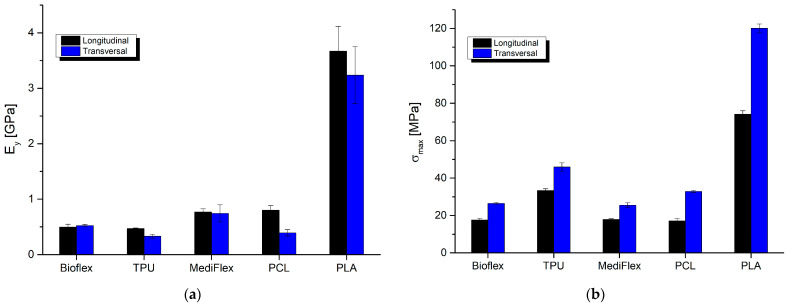
Comparison of longitudinal and transversal results of compression testing of filaments: (**a**) Young modulus; (**b**) maximal stress.

**Figure 11 materials-16-03534-f011:**
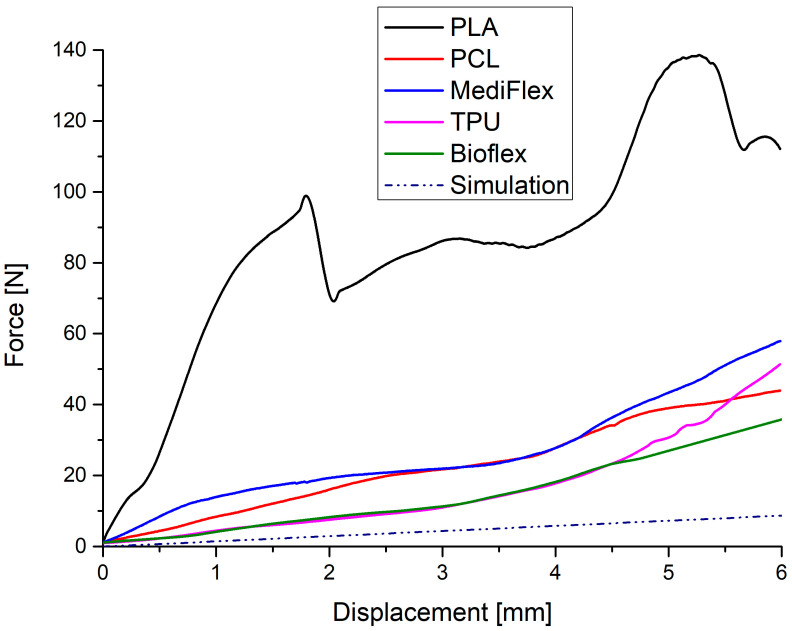
Exemplary bending stress–strain curves for tested filaments and simulated nasal cartilage behavior.

**Figure 12 materials-16-03534-f012:**
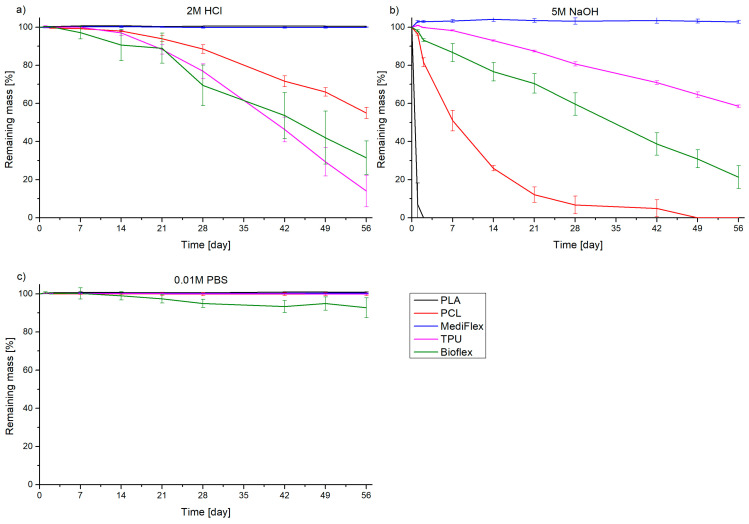
Degradation behavior of filaments in different media: (**a**) 2 M HCl, (**b**) 5 M NaOH, (**c**) 0.01 M PBS.

**Figure 13 materials-16-03534-f013:**
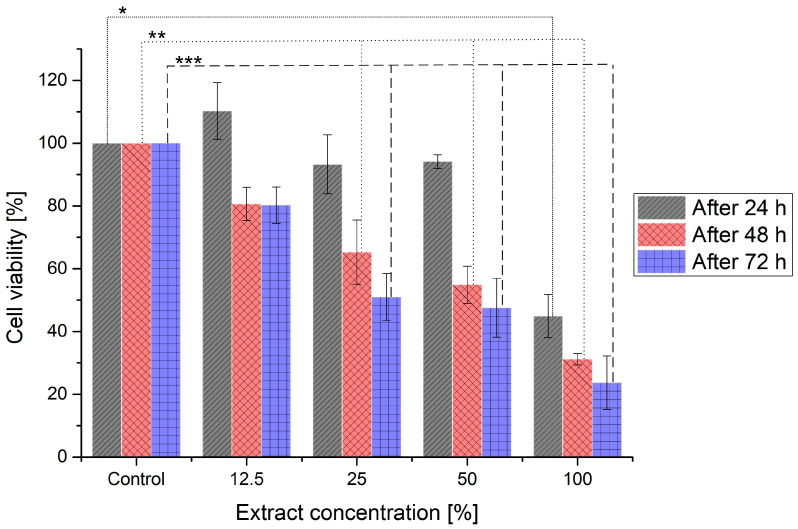
Cell viability of L-929 cells following treatment with different extract concentrations. Viability higher than 80% corresponds to no cytotoxicity, 60–80% to mild cytotoxicity, 40–60% to moderate cytotoxicity and lower than 40% to severe cytotoxicity. *, **, ***—significant differences between the means at the 0.05 level (Tukey test) in comparison to control after 24, 48, and 72 h, respectively.

**Figure 14 materials-16-03534-f014:**
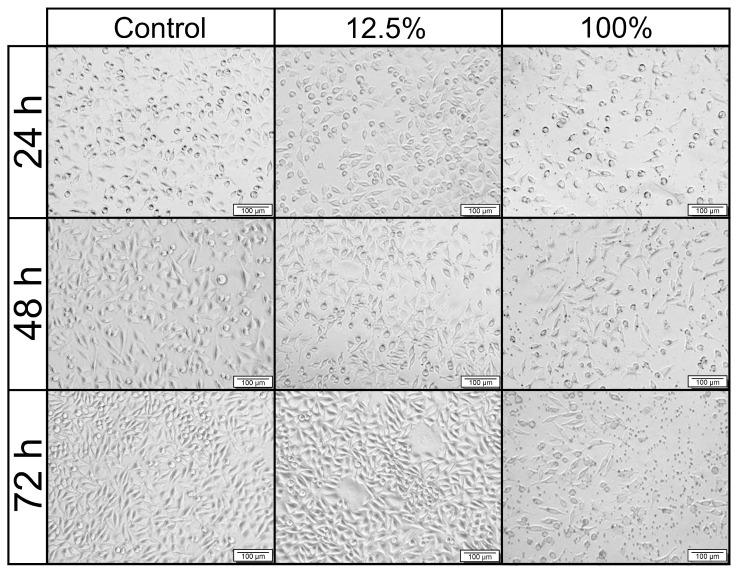
Representative pictures of L-929 cells’ morphology following 24, 48, and 72 h of incubation with extracts at different concentrations. Control: the cells incubated with extract-free medium. Scale bar is 100 µm.

**Table 1 materials-16-03534-t001:** Selected properties of filaments used in the study.

Property	Value				
	TPU [14]	MediFlex [15]	PLA [16]	PCL [17]	Bioflex [18]
Color	Natural	Natural	White	White	White
Density (g/cm^3^)	1.23	0.89	1.24	1.28	1.09
Hardness	55 ShD	96 ShA	NG *	NG	27 ShD
Tensile strength at break (MPa)	44	33	53	20	16
Elongation at break (%)	400	>700	NG	560	800
Melting point (°C)	NG	NG	155–160	58–62	185
Glass transition temperature (°C)	NG	NG	55–60	NG	−70

* NG–not given.

**Table 2 materials-16-03534-t002:** The material properties used for the human cartilage model stress analysis [20,21,22,23].

Material Property	Unit	Value
Young’s modulus	MPa	2.72
Poisson’s ratio	-	0.26
Shear modulus	MPa	1.9
Density	g/cm^3^	1.099
Yield strength	MPa	1
Tensile strength	MPa	1.9

**Table 3 materials-16-03534-t003:** The most important established 3D printing parameters.

Printing Parameter	Value				
	TPU	MediFlex	PLA	PCL	Bioflex
Extrusion temperature (°C)	215	240	215	180	190
Bed temperature (°C)	80	80	60	35	60
Cooling	Fan on; ramp up from 0 (1st layer) to 100% (3rd layer)				
Layer height (mm)	0.15				
First layer height (mm)	0.2				
Wall thickness	3 contours				
Closing layers (top; bottom)	0; 0				
Infill	20%, gyroidal				
Base printing speed (mm/s)	40 (walls), 50 (infill)				
Retraction length (mm)	4				

**Table 4 materials-16-03534-t004:** Stress analysis results in the form of maximal displacement observed under specified load obtained from simulation in Autodesk Inventor 2023.

Bending		Compression	
Force (N)	Max. Displacement (mm)	Force (N)	Max. Displacement (mm)
1	0.6886	10	0.1753
2	1.377	20	0.3506
3	2.066	30	0.526
4	2.754	40	0.7013
5	3.443	50	0.8766
6	4.131	60	1.052
7 *	4.82	70	1.227
8 *	5.509	80	1.403
9 *	6.197	90 *	1.578
10 *	6.886	100 *	1.753
		110 *	1.929
		120 *	2.104
		130 *	2.279
		140 *	2.454

* The analysis finished with a warning: “The deformation is large compared to the model size. Verify that load and constraint settings are appropriately scaled”.

**Table 5 materials-16-03534-t005:** Descriptions of characteristic bands observed during FTIR studies for the studied filaments.

Band	Wavelength (cm^−1^)	Assignment
ν(N-H)	3300	N-H stretching vibration of urethane groups
ν(C-H)	3100–3000	C-H stretching vibrations of aromatic bonds
ν(C-H)	3000–2800	C-H symmetric and asymmetric stretching vibrations of methylene and methyl groups
ν(C=O)	1750–1725	C=O stretching vibration of ester groups
ν(C=O)	1690	C=O stretching vibration of urethane groups
ν(C=C)	1600, 1490, 1450	C=C stretching vibration of aromatic groups
ν(C-N)	1220	C-N stretching vibration of urethane groups
ν(C-O)	1160	C-O stretching vibration of ester groups
ν(C-O)	1100	C-O stretching vibrations of ether groups

**Table 6 materials-16-03534-t006:** DSC peaks description obtained by Proteus80 software.

Filament	Peak No.	Temperature Range (°C)	Energy (J/g*K)
Bioflex	I	104.6–138.8	24.59
TPU	I	205.9–225.1	15.20
MediFlex	-	92.3–99.4	0.7287
PCL	I	54.5–63.7	43.57
	II	75.4–91.3	2.61
PLA	I	58.6–62.3	0.61
	II	108.9–139.2	−14.42
	III	146.2–156.4	17.17

**Table 7 materials-16-03534-t007:** Summary of filaments’ MFR and MVR analysis.

Filament	MFR (g/10 min)	MVR (cm^3^/10 min)
Bioflex	70.20 ± 0.67	52.57 ± 0.50
TPU	51.11 ± 0.93	47.3 ± 1.1
MediFlex	10.25 ± 0.31	11.67 ± 0.30
PCL	37.13 ± 0.81	34.2 ± 1.3
PLA	16.59 ± 0.36	15.01 ± 0.31

**Table 8 materials-16-03534-t008:** Summary of mechanical testing results: Young modulus (E) and maximal stress (σ) of tested filaments in compression testing, both longitudinal (l) and transversal (t) to infill, and maximal force (F_max_) observed in bending test.

Filament	E_yl_ (GPa)	σ_ml_ (MPa)	E_yt_ (GPa)	σ_mt_ (MPa)	F_max_ (N)
Bioflex	0.495 ± 0.052	17.60 ± 0.80	0.541 ± 0.021	26.46 ± 0.45	28.3 ± 6.5
TPU	0.468 ± 0.012	33.36 ± 0.92	0.332 ± 0.035	45.9 ± 2.3	55.3 ± 4.1
MediFlex	0.770 ± 0.056	17.88 ± 0.36	0.74 ± 0.16	25.5 ± 1.4	58.8 ± 7.5
PCL	0.801 ± 0.081	17.2 ± 1.2	0.392 ± 0.058	32.87 ± 0.62	37.9 ± 9.3
PLA	3.67 ± 0.45	74.2 ± 1.8	3.24 ± 0.52	120.1 ± 2.3	145 ± 17

## Data Availability

Data will be available in the Bridge of Data repository.

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
