# Peer review of "Investigation on Filaments for 3D Printing of Nasal Septum Cartilage Implant"

_materials, 2023, doi:10.3390/ma16093534_

Round 1
Reviewer 1 Report
Title of the manuscriptt: Investigation on filaments for 3D printing of nasal septum cartilage implant.
I appreciate the authors and their consistent results.
Few comments are here to be updated.
1. Authors advised to explain more in detailed interconnection of nasal septum reducing the cell viability/ cytotoxicity
2. Authors conducted the contact angle of Bioflex and PLA. If authors explain in more detail of absorption of tissue with fixed time would be more significant.
3. FTIR image results should be presented more specifically in chemical and bonds.
4. Degradation Studies: The temperature of the reaction condition and pH and of the solution also make an impact on degradation, so authors suggested to mention the pH maintained in degradation studies
Author Response
Respected Reviewer,
Thank you for your valuable insight into our manuscript. Your comments have been considered and appropriate corrections have been made to the text. Below, please find the responses (bold) to your comments (italic, original writing).
- Authors advised to explain more in detailed interconnection of nasal septum reducing the cell viability/ cytotoxicity
The biocompatibility of the proposed implants is fundamental for regenerative effect and prevention of inflammatory reactions. Section 3.9 was revised and broadened to include these issues.
- Authors conducted the contact angle of Bioflex and PLA. If authors explain in more detail of absorption of tissue with fixed time would be more significant.
The impact of contact angle on tissue-implant interactions was added to section 3.6.
- FTIR image results should be presented more specifically in chemical and bonds.
For clearer interpretation of FTIR spectra, the table with descriptions of characteristic bands was added (Table 5).
- Degradation Studies: The temperature of the reaction condition and pH and of the solution also make an impact on degradation, so authors suggested to mention the pH maintained in degradation studies
The pH of degradation media was added (Section 2.10).
We hope that the explanations provided are sufficient and meet your highest standards.
Best regards,
Przemysław Gnatowski
Reviewer 2 Report
Introduction - Need a table for 5 different filament physical and chemical properties.
Need a referance for cell line study and Degradation Studies.
Need justification for media selection.
The septal cartilage model was prepared by the segmentation of the MRI scan of the human head - how many MRI scan were used. Avg. size decided with which criteria.
Author Response
Respected Reviewer,
Thank you for your valuable insight into our manuscript. Your comments have been considered and appropriate corrections have been made to the text. Below, please find the responses (bold) to your comments (italic, original writing).
Introduction - Need a table for 5 different filament physical and chemical properties.
The table with properties of tested filaments was added (Table 1.)
Need a referance for cell line study and Degradation Studies.
The cytotoxicity studies were performed according to the ISO-10993:5(2009)E, in which L-929 cell line (adult mouse fibroblast cell line) is recommended for the biological evaluation of biomedical devices – part 5: tests for in vitro cytotoxicity. Therefore, we applied this cell line for the studies concerning cytotoxic activity of our samples. Sections 2.11 and 2.12 were merged, and the explanation about L-929 cell line was added to the text.
Need justification for media selection.
We purchased L-929 cells from MERCK. Supplier recommended to culture these cells in Low Glucose Dulbecco’s modified Eagles’ medium (DMEM) supplemented with 10% fetal bovine serum (FBS), 100 U/mL penicillin, 100 ug/mL streptomycin and 2 mM glutamine. We would like to add that other laboratories used this medium for performing experiments with L-929 cells [Nabavizadeh MR, Moazzami F, Gholami A, Mehrabi V, Ghahramami Y. Cytotoxic effects of nano fast cement and proroot mineral trioxide aggregate in L929 cells fibroblast cells: an in vitro study. J Dent Shiraz Univ Med Sci. March 2022;23(1):13-19]. The mention about supplier recommended medium was added to the text (Section 2.11).
The septal cartilage model was prepared by the segmentation of the MRI scan of the human head - how many MRI scan were used. Avg. size decided with which criteria.
The model was prepared using MRI scan of the human head of one male adult subject without septal deviation (text adjusted in section 2.2.). The procedure was prepared with the idea of patient-specific implants, where doctors after obtaining MRI scans can prepare implants matched to patient needs using this procedure. In the case of large-scale production of ready-to-use implants, the preparation of different sizes of implants would be indeed necessary. However, at this stage of study we felt that it is not essential in comparison to finding proper material.
We hope that the explanations provided are sufficient and meet your highest standards.
Best regards,
Przemysław Gnatowski
Reviewer 3 Report
This is an interesting study about filaments for 3D printing of nasal septum cartilage implant. The paper focused on assessing the possibility of usage of nasal septum cartilage implant 3D printed from various market available filaments. Five different types of laments were used.
The paper is well written. However, some issues remain.
In the Introduction section, the authors freqeuntly confound septoplasty and rhinoplasty concerning technique (e.g., spreader graft are not used in septoplasty but not in rhinoplasty), complications (alar collapse is not a complication of septoplasty). Moreover, fracture of the nasal bones is rarely associated to traumatic deviation of the nasal septum. Please correct it.
The authors should specify that a subject without septal deviation was used as anatomical model.
Stress analyses were reported only for one material. Which was this materials and what about results from the other materials? Similarly, please report cytotoxiticy studies for all the materials.
The discussion is completely lacking.
The authors must better describe how these materials can be used in a septoplasty.
Author Response
Respected Reviewer,
Thank you for your valuable insight into our manuscript. Your comments have been considered and appropriate corrections have been made to the text. Below, please find the responses (bold) to your comments (italic, original writing).
In the Introduction section, the authors freqeuntly confound septoplasty and rhinoplasty concerning technique (e.g., spreader graft are not used in septoplasty but not in rhinoplasty), complications (alar collapse is not a complication of septoplasty). Moreover, fracture of the nasal bones is rarely associated to traumatic deviation of the nasal septum. Please correct it.
According to literature, spreader grafts are used both in septoplasty (10.1097/SCS.0000000000002898, 10.1001/jamafacial.2018.2118, 10.1001/archfaci.2012.173) and rhinoplasty (10.4172/plastic-surgery.1000944, 10.1007/s00405-011-1837-y). Sometimes it is mentioned as rhinoseptoplasty (10.1007/s00266-015-0597-2, 10.1002/lary.28974).
Concerning alar collapse, although manuscript does not state that alar collapse is a complication of septoplasty, there are literature entries identifying nasal valve collapse as a result of failed septoplasty: 10.1186/s40463-019-0394-z, 10.1111/j.1749-4486.2011.02385.x, 10.7759/cureus.33073.
The manuscript text does not state the fracture of nasal bones is associated with deviation of the nasal septum. The intention was to highlight that fracture of the nasal bones can result in nasal septum damage and deformations, which raises the necessity of surgical intervention. The fracture of nasal bones often results in septal fractures (more than 90%: 10.1097/01.PRS.0000096705.64545.69).
The introduction section was adjusted to clarify issues raised.
The authors should specify that a subject without septal deviation was used as anatomical model.
The subject description was added to section 2.2.
Stress analyses were reported only for one material. Which was this materials and what about results from the other materials? Similarly, please report cytotoxiticy studies for all the materials.
Stress analysis was performed for human nasal cartilage and was then compared with mechanical testing results of samples prepared from five different materials, for selection of appropriate material for the implant. The text was adjusted in sections 2.3., 3.2. Suitable mentions were already in captions of mechanical testing figures. The only material which showed a set of suitable properties for usage as the biodegradable septum implant was Bioflex, which offered biodegradability, best mechanical properties similarity to simulated tissue and the lowest water contact angle. Other materials felt short in meeting demands for septum implant, thus were not tested for cytotoxicity.
The discussion is completely lacking.
The Discussion section was combined with Results as guide for authors allows. Each results section contains a discussion part, and that part of manuscript was broadened in sections 3.1., 3.2., 3.6., 3.7. and 3.9. with respect to the total manuscript length.
The authors must better describe how these materials can be used in a septoplasty.
The work focuses on preparation of nasal septum implant, but the results could be used for preparation of other nasal cartilage implants. The manuscript text was adjusted in introduction and conclusion sections to address this comment.
We hope that the explanations provided are sufficient and meet your highest standards.
Best regards,
Przemysław Gnatowski
Round 2
Reviewer 3 Report
Thanks for improving the manuscript.